

# Metagenomes and metagenome-assembled genomes from tidal lagoons at a New York City waterfront park

Sally Kong[1], Eliana Abrams[1], Yehuda Binik[1], Christina Cappelli[1], Mathew Chu[1], Taiyo Cornett[1], Isayah Culbertson[1], Epifania Garcia[1], Jada Henry[2], Kristy Lam[1], D.B. Lampman[1], Grace Morenko[1], Illusion Rivera[2], Tanasia Swift[2], Isabella Torres[2], Rayven Velez[2], Elliot Waxman[1], Serena Wessely[1], Anthony Yuen[1], Casey K. Lardner[1] and J.L. Weissman[3,4]

[1] Genspace, Brooklyn, NY, United States of America
[2] Billion Oyster Project, New York, NY, United States of America
[3] Department of Ecology & Evolution, State University of New York at Stony Brook, Stony Brook, NY, United States of America
[4] Institute for Advanced Computational Science, State University of New York at Stony Brook, Stony Brook, NY, United States of America

Corresponding authors
Casey K. Lardner,
clardner@genspace.org
J.L. Weissman,
jlweissman@gmail.com

## ABSTRACT

New York City parks serve as potential sites of both social and physical climate resilience, but relatively little is known about how microbial organisms and processes contribute to the functioning of these deeply human-impacted ecosystems. We report the sequencing and analysis of 15 shotgun metagenomes, including the reconstruction of 129 high-quality metagenome-assembled genomes, from tidal lagoons and bay water at Bush Terminal Piers Park in Brooklyn, NY sampled from July to September 2024. Our metagenomic database for this site provides an important baseline for ongoing studies of the microbial communities of public parks and waterfront areas in NYC. In particular, we provide rich functional and taxonomic annotations that enable the use of these metagenomes and metagenome-assembled genomes for a wide variety of downstream applications.

## INTRODUCTION

We report the sequencing and analysis of 15 shotgun metagenomes from tidal lagoons and bay water at Bush Terminal Piers Park in Brooklyn, NY from July to September 2024 (Fig. 1). Notably, this waterfront park is an active site of ecological research and restoration by the Billion Oyster Project, an environmental nonprofit whose mission is to restore oyster reefs to New York Harbor through public education initiatives. Billion Oyster Project maintains an active community oyster reef in the innermost of our focal lagoons (*Janis, Birney & Newton, 2016*; *Acquie, 2022*).

Bush Terminal Piers Park is developed on a former brownfield, subject to storm- and sea level rise-related flooding, and is a social and environmental amenity for area residents. In combination with efforts to rezone nearby industrial areas for mixed-use development, the
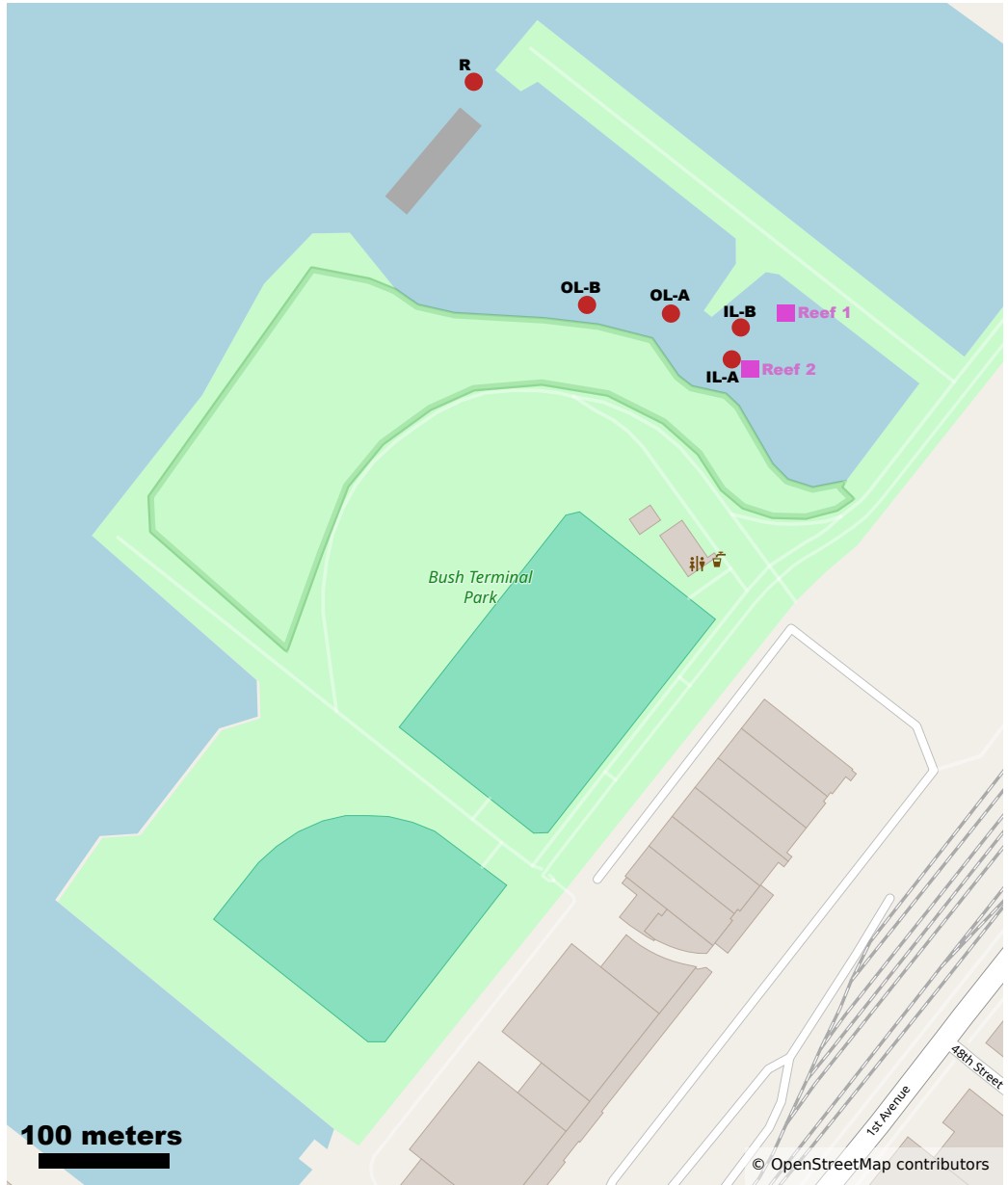

**Figure 1 Map of study site and Bush Terminal Piers Park.** Sample sites and oyster reef locations are noted as red circles and purple squares respectively. Mapping data from OpenStreetMap contributors and used under the Open Database License allowing free adaptation with attribution. ©OpenStreetMap contributors.

area is also impacted by the contested forces of gentrification (*Birney, 2017*). In addition to sports fields, there are a series of short nature trails through a small, wooded area, and walking paths near our focal lagoons, which are used during the summer for community events (*e.g.,* a community boating event in summer 2024).

In aquatic ecosystems, bivalve populations exert strong top-down control on microbial communities *via* size-dependent predation of larger microbes (*Prins, Smaal & Dame, 1997*) and simultaneously redirect nutrients back to these communities through their excretions which are in-turn remineralized by microbes (*Hulot et al., 2020*; *Cherif et al., 2016*), making their impact on community structure hard to predict. Second order effects of bivalve addition, including changes to local hydrology and sedimentation rates, further complicate this picture (*Hulot et al., 2020*). These effects may in turn potentially feedback on oyster population health. In short, it is difficult to predict how the restoration of oyster reefs around New York Harbor will alter local microbial community structure and function. Complicating things further, we do not have a detailed baseline for the microbial community at reef-impacted sites. We constructed a metagenomic time series at this site during mid-to-late summer of 2024 in order to build a location-specific database that will serve as an important resource for future studies of the microbial populations in NYC's waters, particularly at sites of active restoration like Bush Terminal Piers.

## MATERIALS & METHODS

### Sample collection

The City of New York Parks & Recreation granted the approval for this field experiment with project number #768300 to the Billion Oyster Project. Water samples were collected on four sampling dates from July to September 2024 at Bush Terminal Piers Park, Brooklyn, NY, during low tide, when the site forms distinct inner and outer lagoons disconnected from the bay, with the oyster reef located in the inner lagoon (Table 1). Samples were taken from the surface 1m using clean 1L polypropylene bottles after rinsing bottles with sample water three times. Sampling locations are noted in Fig. 1. On the day of collection, water samples were immediately vacuum-filtered onto 0.22 $\mu$m Cellulose Nitrate Filter membranes (Sigma Aldrich GSWP04700). The filter membranes were then stored at $-80\,°C$ until DNA extraction. We also downloaded metadata from nearby NYC Department of Environmental Protection water quality monitoring stations, including salinity, pH, and nutrient measurements during our sampling period (7/16/2024-9/18/2024; *New York City Department of Environmental Protection, 2025*).

### DNA extraction and sequencing

DNA was extracted from the stored filters using the DNEasy PowerWater Kit (14900-100-N; Quigen, Venlo, The Netherlands) following the manufacturer's protocol. Extracted DNA was quantified using the Qubit dsDNA BR Assay Kit (Q32850; Invitrogen, Waltham, MA, USA) and stored at $-20\,°C$. Library preparation was performed using the Rapid Plus DNA Lib Prep Kit for Illumina (RK20208; AB Clonal, Woburn, WA, USA). Samples were then sequenced on the NovaSeq XP platform with 150 bp paired-end sequencing (Illumina, San Diego, CA, USA), generating high-resolution microbial community profiles. On 9/2/24 one additional inner-lagoon sample was prefiltered using an 0.22 $\mu$m Cellulose Nitrate Filter membranes (GSWP04700; Sigma-Aldrich, Burlington, MA, USA) to remove cells and $MgCl_2$ was added to facilitate viral filter-adsorption and then the sample was refiltered again onto a new 0.22 $\mu$m Cellulose Nitrate Filter to enrich potential viral sequences

**Table 1  Sample details.** Sample A15 (IL 09-02-2024), which was enriched for metaviromics (see 'Methods'), not included in this table.

| Sample | Site | Date | Water temperature (C) |
|--------|------|------|----------------------|
| A01 | Inner Lagoon (IL) | 07–19–24 | 28.4 |
| A02 | Inner Lagoon (IL) | 07–19–24 | |
| A03 | Inner Lagoon (IL) | 08–05–24 | 25.0 |
| A04 | Inner Lagoon (IL) | 08–05–24 | |
| A05 | Inner Lagoon (IL) | 08–05–24 | |
| A06 | Bay Water (R) | 09–02–24 | 24.0 |
| A07 | Inner Lagoon (IL) | 09–02–24 | 24.3 |
| A08 | Inner Lagoon (IL) | 09–02–24 | |
| A09 | Outer Lagoon (OL) | 09–02–24 | 24.1 |
| A10 | Outer Lagoon (OL) | 09–02–24 | |
| A11 | Inner Lagoon (IL) | 09–17–24 | 24.3 |
| A12 | Inner Lagoon (IL) | 09–17–24 | |
| A13 | Outer Lagoon (OL) | 09–17–24 | 24.6 |
| A14 | Outer Lagoon (OL) | 09–17–24 | |

(*Lukasik et al., 2000*). Extraction and sequencing were then performed on this sample as above.

## Sequence analysis

Adapters and low-quality reads were trimmed using fastp v0.23.4 with default settings (*Chen et al., 2018*). Reads from each sample (excluding the virus-enriched sample) were assembled using the SPAdes v4.0.0 genome assembler with option ''–meta'' (metaSPAdes; *Nurk et al., 2017*). Coverage of each contig across all samples was calculated using fairy v0.5.7 (*Nurk et al., 2017*). Metagenomic bins were then inferred from bins for each sample, using coverages across all samples, with MetaBAT2 v2.17 with a minimum contig length set to 2 kb (*Kang et al., 2019*). Bin quality was assessed using CheckM2 v1.0.1 (*Chklovski et al., 2023*).

Bins were annotated with prokka v1.14.6 (*Chklovski et al., 2023*) and eggnogmapper v 2.1.12 (*Seemann, 2014*). We predicted the maximum growth rate of each bin using gRodon v 2.4.0 (*Cantalapiedra et al., 2021*). Taxonomy was assigned to each bin using gtdb-tk v2.1.1 (*Weissman, Hou & Fuhrman, 2021*). We used CoverM v0.7.0 to assess bin abundances across samples (*Chaumeil et al., 2022*), and bin relative abundances were mclr transformed using the SPRING v1.0.4 R package (*Aroney et al., 2025*).

We also ran both prokka v1.14.6 (*Seemann, 2014*) (with option metagenome) and gRodon v2.4.0 (*Weissman et al., 2022*) (with option metagenome_v2) to obtain bulk growth rate predictions for each microbial community. We used sylph v0.8.0 for rapid community-level taxonomic profiling (*Shaw & Yu, 2024b*) and the R package vegan v2.6-8 for NMDS analysis (*Oksanen et al., 2024*).

Finally, we attempted to reconstruct viral genomes by first re-assembling all samples (including virus enriched sample) using SPAdes v4.0.0 genome assembler with option ''–metaviral'' (*Antipov et al., 2020*). Viral sequences were then detected using VirSorter2

v2.2.4 (*Guo et al., 2021*) and further assessed for quality using CheckV v1.0.3 (*Nayfach et al., 2021*). Only high-quality viral genomes as assessed by CheckV were retained.

# RESULTS

## Community composition

We sequenced 15 metagenomes at a depth of 8–10 gb per sample (average 9.8 gb). In general, taxonomic abundances (inferred *via* read-based k-mer sketching *Shaw & Yu, 2024b*) across sample dates and sites remained relatively constant (Figs. 2A–2D), though samples tended to group by date and by site within dates in their composition (Fig. 2E). We noted that early-season samples (July, August) had a higher proportion of *Rhodobacteriales*, whereas later season samples (September) tended to have a higher proportion of *Pelagibacteriales* and *Flavobacteriales* (Fig. 2C). One sample, the lone sample taken from site "R" representing water sampled directly from the shore of the Upper New York Bay directly outside the inlet to the outer lagoon, rather than from either tidal lagoon, had a distinct taxonomic composition with a higher proportion of *Pelagibacteriales* and a low proportion of both *Rhodobacteriales* and *Flavobacteriales*.

## Reconstructed bins

We obtained 1,016 total bins, 129 of which were determined to be high quality with less than 5% contamination and being over 90% complete with the total number of contigs ranging from 8–692 and the average contig length ranging from 4,785–372, 700 bp (Table S1; *Bowers et al., 2017*). Another 366 were determined to be of medium quality (<10% contamination, >50% completeness). All bins have annotations, including trait data, but we restrict our discussion of results to our high-quality bins. Our high-quality bins span 10 phyla and at least 45 genera. A total of 16 high-quality bins could not be confidently assigned to a known genus and our lone bin from the *Chlamydiota* could not be assigned to a known family, potentially representing novel diversity at these taxonomic levels.

## Trait data

These bins have diverse functional content on the basis of assigned gene families, with bins from the same phylum typically having a similar number of functional gene assignments but with a great deal of variation both within and between phyla (Fig. 3). Notably, our bins span a range of growth classes, including slow-growth classes that are often missed by isolation-based methods (Figs. 4A–4B; *Weissman, Hou & Fuhrman, 2021*).

Community-wide average maximum growth rate predictions varied across sample sites, with inner lagoon samples seeming to have higher growth rates, though our sample sizes were insufficient to detect any significant effect of sample site on growth (Figs. 4C–4D; ANOVA, $p > 0.39$, $df = 2$, $F = 0.999$). Looking across inner-lagoon samples, for which we had the most data, the relative abundance of bins was correlated with that bin's codon usage bias, which is the basis for our genomic maximum growth rate predictions, indicating that increased genomic growth optimization is correlated with higher relative abundances in these samples (Fig. 4E; linear regression, $p < 1e - 16$, adjusted $r^2 = 0.166$, coefficient = 5.97).

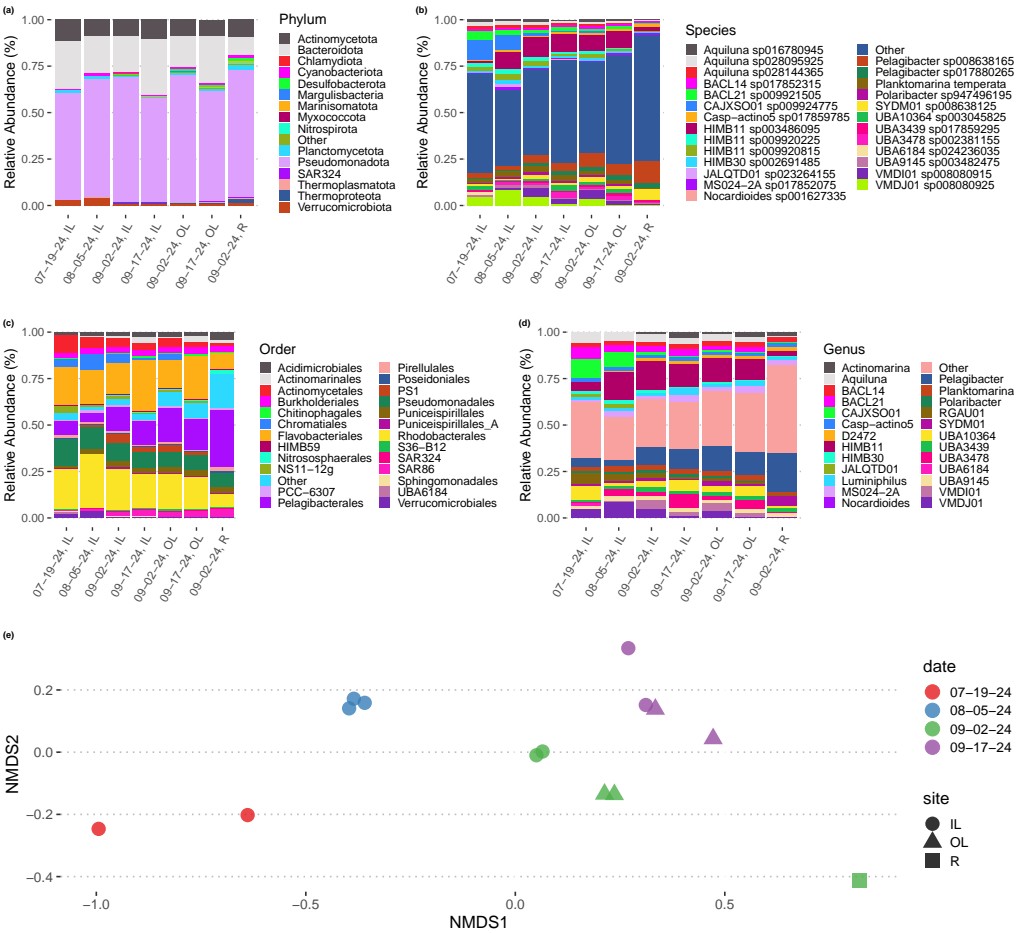

**Figure 2** **Taxonomic composition of tidal lagoons over the course of a summer.** (A–D) Relative abundance of taxonomic groups in each in each site during each day (averaged over replicates) at various levels of taxonomic resolution. (E) Two dimensional non-metric multidimensional scaling plot of species-level taxonomic composition across our samples groups sampled by site and date. Taxonomic composition inferred directly from reads by sylph (*Chaumeil et al., 2022*).

## Viruses

We recovered 50 high-quality viral metagenome-assembled genomes (vMAGs; checkV quality classification). Of these, 26% were assembled from our virus-enrichment treated sample (see Methods). Of these viruses, five were predicted to be single-stranded DNA viruses and the remainder were predicted to be double-stranded.

## DISCUSSION

We present a comprehensive baseline metagenomic dataset for the urban tidal lagoons located at Bush Terminal Piers Park in Brooklyn, NY, including 15 shotgun metagenomes and 129 high-quality metagenome-assembled genomes (MAGs) with rich functional and taxonomic annotations. Our efforts supplement existing microbiome datasets from the NYC subway system, wastewaters, and park soils (*Afshinnekoo et al., 2015*; *Gulino et al.,*

PeerJ

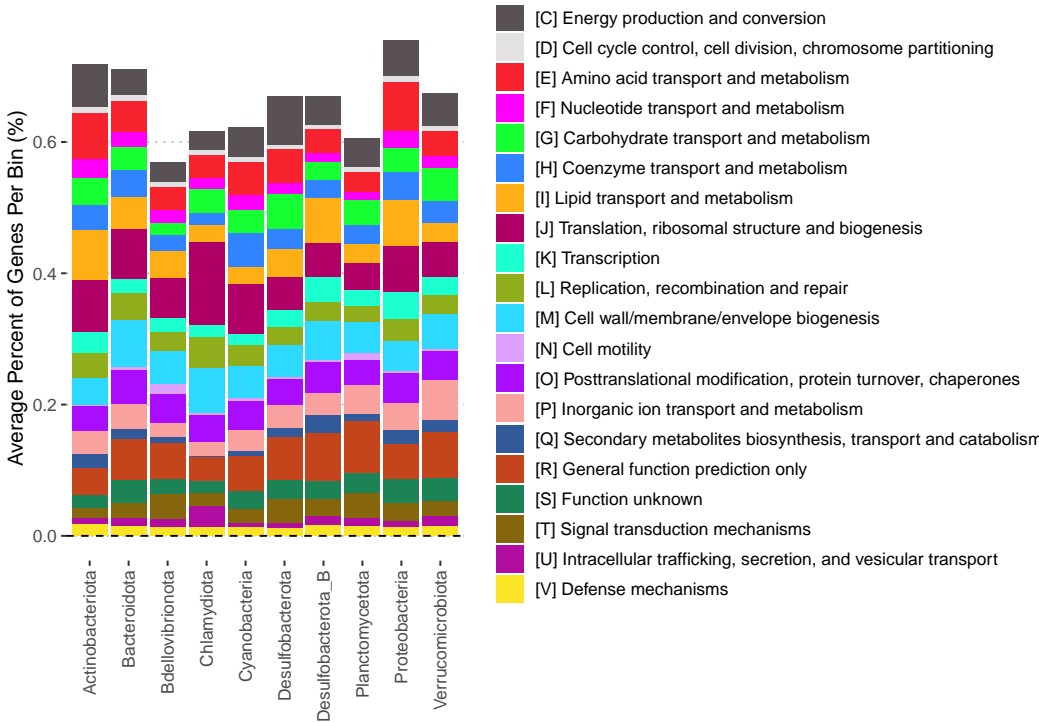

**Figure 3 Functional content of high-quality metagenomic bins.** Each bar represents the average percent of genes belonging to a particular functional class across bins in each phylum. Functional classifications given by eggnogmapper (*Shaw & Yu, 2024a*) and bracketed letters specify the functional family classifications given by eggnogmapper in reference to the Clusters of Orthologous Genes (COG) database.

*2020*). Our focus on waterfront parks and their aquatic microbiomes centers the unique vulnerability of waterfront spaces in a coastal city exposed to increasingly severe flooding (*Rosenzweig et al., 2024*). More broadly our work complements a growing body of research examining the functional capacity of microbiomes in human-constructed spaces and their potential impacts on human wellbeing (*National Academies of Sciences, Engineering, and Medicine, 2017*; *Charlop-Powers et al., 2016*; *Bruno et al., 2022*; *Ryon et al., 2022*; *Mason et al., 2016*).

Our community-level data revealed overwhelmingly stable taxonomic composition despite daily flushing by the tides (Fig. 2E), with a pattern of gradual taxonomic succession over the course of the season. In comparison to water sampled directly from the Upper Bay of New York, both tidal lagoons had distinct taxonomic patterns. Stable differentiation between the lagoons and surrounding waters despite flooding with each tide suggests that either (1) the local environment quickly seeds microbes into these waters (*e.g.*, from the surrounding sediments; *Lennon & Jones, 2011*), or (2) by the time of sampling at low-tide the microbial communities in these waters have responded to changes in local conditions in a predictable diel pattern (*e.g.*, shallower, stagnant conditions with abundant invertebrates present including oysters and crabs; *Zhao et al., 2023*; *Becker et al., 2020*). We expect the reality to be some combination of the two. In contrast, we did not see any directional

Kong et al. (2025), *PeerJ*, DOI 10.7717/peerj.20081 7/15

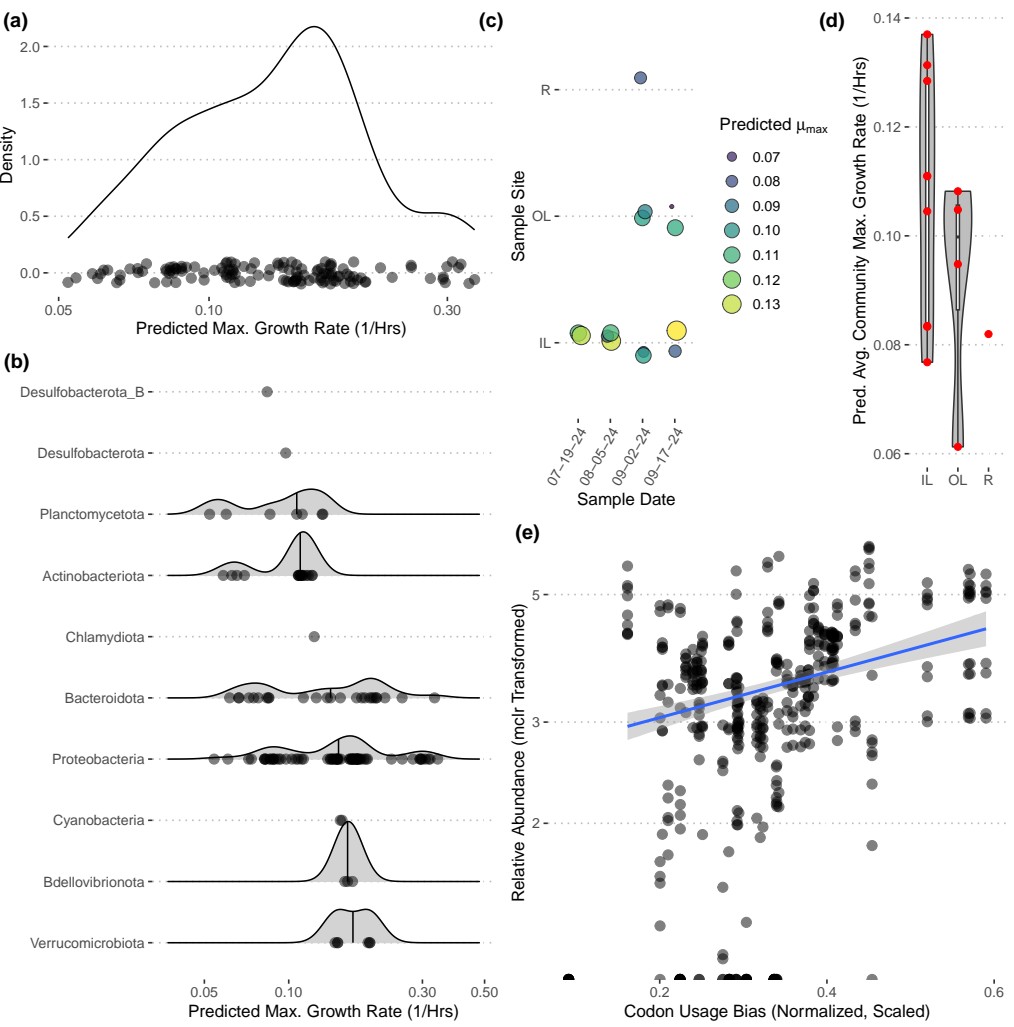

**Figure 4 Predicted maximum growth rates for metagenomes and metagenomic bins.** (A–B) Distribution of predicted maximum growth rates for metagenomic bins assuming a reference temperature of 25 °C. (C–D) Predicted community-wide average maximum growth rates for each metagenomic sample. Jitter added in panel (C). (E) The relative abundances of individual bins across inner-lagoon samples show a positive association with the codon usage bias of each bin. All growth rates and codon usage bias inferred using gRodon (*Kang et al., 2019*; *Weissman, Hou & Fuhrman, 2021*).

pattern of succession over time in our community-level maximum growth rate predictions (Figs. 3C–3D), although there may be differentiation across sample sites (not significant, ANOVA, $F = 0.999$).

Our reconstructed MAGs had diverse taxonomic affiliations and functional content. Notably, 16 of our MAGs could not be classified at the genus level to anything in the GTDB v220 taxonomy. These MAGs had a range of predicted growth rates that suggested many would not have been readily captured by short-term culturing approaches that often miss slow-growing organisms (maximum growth rates greater than 0.13 in Fig. 4A, corresponding to minimum doubling times longer than 5 h; *Weissman, Hou & Fuhrman,*

*2021*). We also captured MAGs that ranged widely in their abundances across samples, with fast-growing MAGs predicted to have the highest relative abundances on average (Fig. 4E). These MAGs varied greatly in the proportion of their coding genome associated with particular functions (Fig. 3), suggesting that this library covers a range of ecological niches.

## CONCLUSIONS

As a dense, coastal city, NYC serves as a valuable model for understanding how climate change-related extreme weather events and sea level rise will impact complex socio-ecological systems (*Rosenzweig et al., 2024*). In particular, New York parks serve as potential sites of both social and physical climate resilience, providing relief from recurring heatwaves and flooding events at the same time they allow for community organizing in areas that have suffered a historic lack of investment (*Rosan, 2012*; *Fainstein, 2018*; *Jabareen, 2014*; *NYC Mayor's Office of Climate & Environmental Justice, 2023*). Our metagenomic database for this site provides an important baseline for ongoing studies of the microbial communities of New York City's parks and waterfront areas.

## ACKNOWLEDGEMENTS

Portions of this text were previously published as part of a preprint (*Weissman et al., 2025*).

### Funding

This work was supported by a community biology grant from Experiment.com and a community science grant from Con Edison, both to Genspace. The funders had no role in study design, data collection and analysis, decision to publish, or preparation of the manuscript.

### Grant Disclosures

The following grant information was disclosed by the authors:
Experiment.com.
Con Edison.

### Competing Interests

Jada Henry, Illusion Rivera, Tanasia Swift, Isabella Torres, and Rayven Velez are employees of the Billion Oyster Project. Serena Wessely and Casey K Lardner are employees of Genspace. Sally Kong, Eliana Abrams, Yehuda Binik, Christina Cappelli, Mathew Chu, Taiyo Cornett, Isayah Culbertson, Epifania Garcia, Kristy Lam, D B Lampman, Grace Morenko, Elliot Waxman, and Anthony Yuen are members of Genspace.

### Author Contributions

- Sally Kong conceived and designed the experiments, performed the experiments, analyzed the data, prepared figures and/or tables, authored or reviewed drafts of the article, and approved the final draft.

- Eliana Abrams performed the experiments, analyzed the data, authored or reviewed drafts of the article, and approved the final draft.
- Yehuda Binik performed the experiments, analyzed the data, authored or reviewed drafts of the article, and approved the final draft.
- Christina Cappelli performed the experiments, analyzed the data, authored or reviewed drafts of the article, and approved the final draft.
- Mathew Chu performed the experiments, analyzed the data, authored or reviewed drafts of the article, and approved the final draft.
- Taiyo Cornett performed the experiments, analyzed the data, authored or reviewed drafts of the article, and approved the final draft.
- Isayah Culbertson performed the experiments, analyzed the data, authored or reviewed drafts of the article, and approved the final draft.
- Epifania Garcia performed the experiments, analyzed the data, authored or reviewed drafts of the article, and approved the final draft.
- Jada Henry performed the experiments, authored or reviewed drafts of the article, and approved the final draft.
- Kristy Lam performed the experiments, analyzed the data, authored or reviewed drafts of the article, and approved the final draft.
- D.B. Lampman performed the experiments, analyzed the data, authored or reviewed drafts of the article, and approved the final draft.
- Grace Morenko performed the experiments, analyzed the data, authored or reviewed drafts of the article, and approved the final draft.
- Illusion Rivera performed the experiments, authored or reviewed drafts of the article, and approved the final draft.
- Tanasia Swift conceived and designed the experiments, performed the experiments, authored or reviewed drafts of the article, and approved the final draft.
- Isabella Torres performed the experiments, authored or reviewed drafts of the article, and approved the final draft.
- Rayven Velez performed the experiments, authored or reviewed drafts of the article, and approved the final draft.
- Elliot Waxman performed the experiments, analyzed the data, authored or reviewed drafts of the article, and approved the final draft.
- Serena Wessely conceived and designed the experiments, performed the experiments, analyzed the data, authored or reviewed drafts of the article, and approved the final draft.
- Anthony Yuen performed the experiments, analyzed the data, authored or reviewed drafts of the article, and approved the final draft.
- Casey K. Lardner conceived and designed the experiments, performed the experiments, analyzed the data, authored or reviewed drafts of the article, and approved the final draft.
- J.L. Weissman conceived and designed the experiments, performed the experiments, analyzed the data, prepared figures and/or tables, authored or reviewed drafts of the article, and approved the final draft.

## Data Availability

The metagenomes are available in SRA: PRJNA1251010.

High-quality bins with annotations and code to generate figures are available at Zenodo: Weissman, J., Kong, S., Cappelli, C., Chu, M., Cornett, T., Culbertson, I., Garcia, E., Lampman, D., Morenko, G., Wessely, S., Yuen, A., & Lardner, C. (2025). Metagenomes and Metagenome-Assembled Genomes from Tidal Lagoons at a New York City Waterfront Park [Data set]. Zenodo. Available at https://doi.org/10.5281/zenodo.16771119.

The scripts to run the metagenomic analysis are available at Github and Zenodo:

– Available at https://github.com/jlw-ecoevo/bushterminalnyc-metagenomics.

– Weissman, J., Kong, S., Cappelli, C., Chu, M., Cornett, T., Culbertson, I., Garcia, E., Lampman, D., Morenko, G., Wessely, S., Yuen, A., & Lardner, C. (2025). Metagenomes and Metagenome-Assembled Genomes from Tidal Lagoons at a New York City Waterfront Park [Data set]. Zenodo. Available at https://doi.org/10.5281/zenodo.16771119.

## Supplemental Information

Supplemental information for this article can be found online at http://dx.doi.org/10.7717/peerj.20081#supplemental-information.

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
