# Peer review of "Metagenomes and metagenome-assembled genomes from tidal lagoons at a New York City waterfront park"

_PeerJ, doi:10.7717/peerj.20081_

## Round 0.1 · original submission · Major Revisions

The manuscript has not been sent out for external review. Several issues should be addressed first (see below).

Regards,
Michael

Line 25. Abstract should start with a description of the field of study and a statement of the unmet need.
Line 27. fix typo
Line 137. Discussion should provide context within the scientific literature. As submitted, there are no citations in this section. This makes it difficult for the reader to understand the importance of the results and it makes it hard for me to find qualified reviewers..

---

## Round 0.2 · Minor Revisions

We received three positive reviews but two reviewers identified minor issues, which should be addressed. Please also address the issues below.

Michael

The lack of metadata is concerning. Did you collect salinity, nutrients..? Can you predict ranges from public datasets? For example, the city provides public access to NYC harbor water quality data: https://data.cityofnewyork.us/Environment/Harbor-Water-Quality/5uug-f49n/about_data

The reviewer’s comments about viruses are insightful. Did you consider using metaviral spades for the sample that was prefiltered and spiked with Mg to enrich for viruses?

Line 191. Add period.
Line 208. Format references consistently. Here only cap proper nouns in article titles. See also lines 228, 240, 243…

·

Basic reporting

Overall, the paper is clear and well presented. My comments here are minor:

1. There are compression artifacts in the figures. I am unsure if this arose from the creation of the pdf at the submission portal...but just re-confirm the quality of your image files.

2. For Table 1, I am unsure why there is an empty "notes" column.

3. Figure 1: not sure if this is an artifact of the article submission process, but the very top of Fig 1 (specifically the word “phylum”) is cut off.

4. Figure 1: Is this figure showing the taxonomic composition of the high-quality bins? Or is this read-based taxonomic mapping? This information is not given in the figure legend or the main text (and should be included in both locations).

Experimental design

1. More details need to be provided on the sample collection methods. Where within the lagoons (and then out in the bay) were these samples collected? Was it surface waters? Near the oyster reefs? Additionally, how was the water sample collection performed?

2. It would be useful to include a map that highlights the locations of the sampling.

3. For your sequencing: include a brief description of the library preparation kit used. Also, what was your sequencing read length? What was your sequencing depth/number of reads generated?

4. Line 81: You mention a viral enrichment protocol but then do not report on any viral genomes. Were there viral genomes in the dataset? This should be mentioned briefly. Also, for this enrichment protocol, I am confused about the "pre-filtration" step. Was this a filtration step prior to filtering the water through the 0.22 um filter? What type and size of filter was used?

Validity of the findings

no comment

Reviewer 2 ·

Basic reporting

The manuscript is well written and the authors clearly explain the purpose of the paper. The figures are good and self-explanatory and fit with the overall message of the manuscript.

Experimental design

Both methods and data analysis are appropriate for extracting biological information about microbial communities.

Validity of the findings

Due to the field of research the biodiversity revealed in the manuscript, as well as, the assembled MAGs are novel. It is noteworthy that the authors both in the Conclusions and throughout the manuscript do not exaggerate the value of the findings and clearly state that baseline generation was what they did.

Additional comments

I consider that this is a good exploratory work and that its main value is the generation of a baseline for a system with anthropogenic impact. As such, it does not test any hypotheses, but it will be useful for future research in the same system.

·

Basic reporting

Pass

Experimental design

Pass

Validity of the findings

Pass

Additional comments

These comments were also made in the manuscript:

Line 50: Awkward phrasing, and it was established earlier in the paragraph that this park sees a great deal of residental traffic. Perhaps combine the first half of this sentence with the last? "...lagoons, which are used frequently during the summer for community events."

Lines 109-110: Why was this site chosen and compared to the lagoon time series? Should be included in the methods section

Line 173: Replace "of" with "a"

Table 1: Notes column unnecessary, consider removing

Figure 1: Would it be worth averaging the relative abundance values of the samples? They were taken on the same day (either IL or OL) and presumably taken at the same location. If so, that would remove the "(AXX)" text on the x-axis and reduce clutter. Averaging values would also make the figure more easy to interpret

Would it also be worth rearranging the values by sample location then date? There might be location-specific differences in taxonomic abundance worth representing more clearly and making mention of in the Results/Discussion. For instance, for Figure 1a, it appears that Desulfobacterota are more prevalent in R and OL than in IL

Figure 1a also looks slightly cut off at the top

Figure 2: What is the purpose of the [Letter] in the legend?

Figure 3 Legend: "o" is missing in "oC"

Figure 3: I don't understand (for figure 3c) the range of values relative to their colors and how they are plotted relative to the grey dotted line

For instance, hotter colors seem to correspond to larger circles and higher umax, so what does yellow represent?

I'm guessing the grey dotted line is just a reference point and the values of each sample are jittered, correct?

---

## Round 0.3 · accepted · Accept

The two reviews recommended acceptance without qualifications. I agree with them. My decision is accept. I did notice two typos (see below) that should be corrected before the manuscript is ready for publication.

Regards,

Michael
- line 88. Subscript "2" in MgCl2
- line 324. "t" is missing in "tomorrow"

·

Basic reporting

I have no further comments.

Experimental design

I have no further comments.

Validity of the findings

I have no further comments.

·

Basic reporting

No comment.

Experimental design

No comment.

Validity of the findings

No comment.

Additional comments

The authors present new and important work regarding microbial community composition and growth rates over time in a highly urbanized ecosystem. They also demonstrate their objectives and findings clearly and concisely. Reviewer comments on both the manuscript and the figures were also fully addressed.